# The Adsorption of Small Molecules on the Copper Paddle-Wheel: Influence of the Multi-Reference Ground State

**DOI:** 10.3390/molecules27030912

**Published:** 2022-01-28

**Authors:** Marjan Krstić, Karin Fink, Dmitry I. Sharapa

**Affiliations:** 1Institute for Theoretical Solid State Physics (TFP), Karlsruhe Institute of Technology (KIT), Wolfgang-Gaede-Str. 1, 76131 Karlsruhe, Germany; marjan.krstic@kit.edu; 2Institute of Nanotechnology (INT), Karlsruhe Institute of Technology (KIT), Hermann-von-Helmholtz-Platz 1, 76344 Eggenstein-Leopoldshafen, Germany; karin.fink@kit.edu; 3Institute of Catalysis Research and Technology (IKFT), Karlsruhe Institute of Technology (KIT), Hermann-von-Helmholtz-Platz 1, 76344 Eggenstein-Leopoldshafen, Germany

**Keywords:** copper paddle-wheel, HKUST-1, MOF, educts adsorption, binding energies, DFT, single-reference methods, multi-reference system

## Abstract

We report a theoretical study of the adsorption of a set of small molecules (C_2_H_2_, CO, CO_2_, O_2_, H_2_O, CH_3_OH, C_2_H_5_OH) on the metal centers of the “copper paddle-wheel”—a key structural motif of many MOFs. A systematic comparison between DFT of different rungs, single-reference post-HF methods (MP2, SOS–MP2, MP3, DLPNO–CCSD(T)), and multi-reference approaches (CASSCF, DCD–CAS(2), NEVPT2) is performed in order to find a methodology that correctly describes the complicated electronic structure of paddle-wheel structure together with a reasonable description of non-covalent interactions. Apart from comparison with literature data (experimental values wherever possible), benchmark calculations with DLPNO–MR–CCSD were also performed. Despite tested methods show qualitative agreement in the majority of cases, we showed and discussed reasons for quantitative differences as well as more fundamental problems of specific cases.

## 1. Introduction

The correct theoretical prediction of “binding energies” is important for many applications, in particular in catalysis, biochemistry and other fundamental and applied fields [1]. However, these calculations are often quite challenging.

In the “molecular field”, investigation of [hetero]dimers of some small-to-middle size organic or inorganic molecules (mainly closed shell single-reference methods), where a significant role is played by the vdW interaction, was the driving force of method development for many decades. This class of molecules can be represented by (H_2_O)_2_ [2], benzene-dimer [3], interactions between buckyballs [4,5] and with buckycatchers [6,7], etc. From the side of “correct answer for correct reason”, much work was performed by P. Hobza [8,9,10] and G. Martin [11,12,13] (in particular MP2.5 [14] and MP2.X [15] approaches took their place in the tool-box of computational chemists), while from the side of “reasonable answer at the low cost” a great advancement in theory is connected to Grimme’s dispersive corrections [16]. Nowadays, a set of dispersion corrections (D2-D3 [17]-D3BJ [18]-D4 [19,20], TS [21], XDM [22,23], dDsC [24,25], NL(VV10) [26,27] and vdW-DF of different order [28,29]) is available to the research community. There are strong debates about the empirical nature of some of those corrections, reasonability of electronic structure in dispersion-corrected minima and other related issues. The theoretical approach of mixing the DFT, which is barely aware of dispersion, with MP2, which strongly overrates it, has had just limited success: further development gave us quite sophisticated approaches where the ratio of spin-components in MP2 is tuned, DFT is admixed and empirical correction is still needed on top of it (DSD–BLYP–NL [30,31], DOD–PBEhB95–D3BJ [32], DOD–SCAN–D4 [33], etc [34]). Alternative approaches such as DFT–SAPT [35,36,37] and ACFD/ALDA [38] despite significant development are still significantly less popular than DFT-D methods (partially because of higher complexity and computational requirements). The DLPNO approach brought research in this field to a new level, making results of CCSD(T) quality accessible for very large systems [39,40,41,42].

Unfortunately, periodic boundary conditions put significant limits on usage of advanced methods listed previously. Thus, quantum chemistry methods for the description of periodic bulk materials and surfaces are mainly limited to dispersion correction and highly parametrized (DFT) functionals [43].

On the overlap of these periodic and non-periodic fields lies a subgroup of modern and perspective types of coordinated porous molecular materials such as zeolites and metal-organic frameworks (MOFs). In particular, MOFs are of great interest nowadays on the point of their catalytic properties, which are quite often related to adsorption of educts (small molecules) on the coordinatively unsaturated centres (CUS) [44]. For example, we recently investigated CO oxidation taking place in HKUST-1, one of the most classical MOFs that has CUSs [45]. At the same time, adsorption of small molecules (i.e., alcohols) on the copper paddle-wheel of Cu_2_(BDC)_2_SURMOF is a promising technique in building sensors of new generation [46]. In general, dicopper tetracarboxylate paddle-wheel (Figure 1) is a key structural motif of many MOFs: Cu_3_(BTC)_2_ (more commonly known as HKUST-1 [47], also known as MOF-199 [48]), Cu-BDC [49], Cu_4_(TDHB) (BUT-155) [50], Cu_2_(EBTC) (MOF-505) [51], etc., as well as of some giant supramolecules (i.e., MOP-1 [52]). Consequently, prediction of adsorption energies of small molecules on this secondary building unit is of great practical interest [53]. Indeed, there are already a few studies in this direction [54,55,56,57,58,59,60,61,62] performed mainly on the DFT level of theory, and in a few cases are combined with different ab initio approaches [63,64,65]. However, working with metal-organic compounds, one should always be aware of possible effects of multi-reference and, associated with that, issues on applicability of different methods. In fact, copper paddle-wheel (PW) is known to be affected by this issue: apart from having almost degenerated singlet and triplet states, its ground state, singlet, cannot be described by a single configuration of the electrons. This obviously casts a shadow on any results obtained with DFT approaches. The analysis of Fractional Occupation Density (FOD) [66,67], one of most reliable criteria for applicability of DFT to multireference systems, shows strong delocalization of FOD over the system (Figure 1). Furthermore, authors are not aware of any work where Complete-Active-Space (CAS) methods were systematically tested on the point of prediction of accurate binding energies.

The possibility to compare results of high-level, single-reference methods with the multi-reference approaches, with DFT and with available experimental data, makes the PW an outstanding object to estimate the quality of different approaches (and to find out the most cost-effective one). Herein, we present our results.

## 2. Methodology

### 2.1. Choice of Methods

To make things practical and not be drowned in a huge variety of DFT-functionals, we limited our DFT calculations to four functionals of different classes that are most wide-spread in the theoretical community: PBE [68,69] (GGA), B3LYP [70] (hybrid), M06–L [71] (local meta–GGA) and BEEF–vdW [72]. The first two functionals we combined with Grimme’s D3 dispersion correction [17] with Becke–Johnson damping [18]. On the side of ab initio methods we looked separately on single- and multi-reference approaches. Among the multi-reference ones, we used CASSCF (that one might expect to underrate the binding energy), NEVPT2 [73,74,75] (as a method superior to CASPT2) and quite novel DCD–CAS(2) [76]. Single-reference approaches are represented by SOS–MP2 [77] that is known to slightly overestimate non-covalent interactions in comparison to canonical MP2 [4] and DLPNO–CCSD without and with triples correction. The latter can be accounted as an “accessible version” of the “golden standard”. Finally, we performed a few benchmark calculations with a computationally expensive DLPNO–MR–CCSD approach (DLPNO–MkCCSD in the notation of developers [78,79,80,81]). All these calculations were performed as single-points on PBE-optimized molecular geometries (Figure 2).

The Gaussian16 [82] was used for all DFT calculations except BEEF–vdW; VASP5 [83,84] was used for BEEF–vdW calculations. All non–DFT calculations (single and multireference post–HF) were performed in ORCA4.2.1 [85,86,87].

We chose dicopper tetraacetate as the model to include the effect of substituents. However, the computational complexity of DLPNO-MR-CCSD did not allow us to perform calculations on that model system, and additional calculations of dicopper tetraformate with CO and ethylene adsorbates were performed.

All above counted calculations except BEEF–vdW were performed with the Ahlrichs def2–TZVP [88] basis set. It was shown that further increase in the basis set did not improve the result for more than 0.2 kJ/mol [54]. Numerical grids for the integration were Grid6 in ORCA and default UltraFine grid in Gaussian16 codes. TightSCF was used in all Orca calculations and TightPNO in all DLPNO calculations, as it was shown to be obligatory for proper description of noncovalent interactions. All MP calculations were performed with frozen core. We investigated the applicability of this approach on test cases and found virtually no effect on relative energy from this approximation.

For all multi-reference calculations (except O_2_ case) a minimal active space of 2 × 2 was applied, but even in this case DLPNO-MR-CCSD benchmarks were extremely expensive, due to the fact that active space orbitals are not localized, and in the particular system active orbitals are metal-centered. To demonstrate the effect of this, we performed additional CAS (2 × 2) calculations of 2 specially designed systems that have comparable size/atomic composition: (1) bis-μ-oxo Cu_2_O_2_(OH)_2_(NH_3_)_2_ with active orbitals on Cu atoms and (2) Cu–C≡C–Cu + *m*-benzyne (C_6_H_4_) with active orbitals localized on the organic part. The second calculation appears to be an order of magnitude faster than the first one (see Appendix A).

Bias correction of second order (bc2) was tested for DCD–CAS(2) and showed no change in results, thus not discussed in the manuscript. Obsolete MRMP2 was tested and showed stability issues, thus not discussed. In all multireference calculations “dissociated complex” was represented by its components optimized individually on PBE–D3BJ and placed in a single input on the distance of 50 Angstroms.

BEEF–vdW calculations were performed with an energy cutoff of 450 eV. Cluster model (cubic cell with lattice vectors 30 Å) both with tetraformate and tetraacetate showed results consistent with other methods for the first adsorption but unreasonably low second adsorption energy. For this reason, as well as for the sake of computational cost, the periodic structure of Cu–BDC (can be also determined as SURMOF–2) was optimized and used. The difference in the first adsorption energy between the cluster and periodic model is negligible. We would like to point out that, despite the presence of BEEF in LibXC and the possibility to use LibXC in ORCA, Turbomole and other computational codes currently correct BEEF–vdW and cannot be performed in those codes, while the nonlocal vdW term is missing.

Gaussian DFT calculations were performed both with low and high spin states (broken symmetry solution was requested for low spin, and stability of wavefunction was controlled). In all cases the low spin solution (singlet for all the cases without O_2_) was found to be energetically lowest, and these results are shown and discussed in the paper. Total energies (also for high-spin state) can be found in Appendix A.

### 2.2. Choice of Adsorbates

For our research we selected seven small molecules that often have a place in HKUST-1–related studies: H_2_O, CO, CO_2_, O_2_, C_2_H_4_, CH_3_OH and C_2_H_5_OH. We investigated their adsorption from one as well as from both apical positions of paddle-wheel structural motif (that could be related to low and high concentrations of adsorbates). Additionally, although CO is principally an ambidentate ligand, we studied “normal”, more stable adsorption through C atom. In total, we present 14 systems (Figure 2).

## 3. Results and Discussion

### 3.1. Dicopper Tetraformate

Figure 3 and Appendix A show the performance of different methods on the smallest systems of our study (CO and ethylene adsorbed on tetraformate system). It is important to mention that exchange of H substituents to Me results in an insignificant decrease in binding energy (on average around 0.03 eV), which justifies our decision to use smaller model systems for extremely computationally expensive DLPNO–MR–CCSD.

First of all, one can see that results of C_2_H_4_ adsorption (Figure 3) are qualitatively consistent: all values lie in a range of 62–147% from experimental values. Furthermore, results of CASSCF simulations show the insignificant difference when compared to HF, while NEVPT2 binding energies are close to that of the MP2 method. Similar results can also be observed for other systems and can be explained in the way that the application of multiple electronic configurations does not improve the quality of the general approach. In fact, both of these approaches yield higher differences with experimental or benchmark values than DFT. As one could expect, MP3 improves the result of MP2, by decreasing the binding energy. As well, SOS–MP2 fixes this overestimation reaching DLPNO–CCSD(T) quality. Keeping this in mind, since SOS–MP2 can be implemented with the cost of N^4^ (N—number of basis functions), it looks for us as one of the most cost-effective approaches. We also speculate that usage of spin-bias (i.e., SOS) might generally improve the quality (and reduce cost) of CASPT2/NEVPT2 approaches with respect to “binding energy” problems (despite making it more empirical). However, the development and implementation of such methodology is beyond the scope of this project. In all studied cases, DCD–CAS(2) is relatively close to the NEVPT2 method. DLPNO–MR–CCSD value (for singlet state) is just 0.02 eV lower than DLPNO–CCSD (for triplet), and such a difference should not be used for deriving any conclusions as it is significantly below experimental accuracy. DFT results show little variability and are all in reasonable agreement with experimental or benchmark data. Overall, adsorption of the first ethylene molecule agrees well with the experimentally found value, meaning that adsorption on opposite sides of PW is quite independent. Further confirmations of this can be found in Table 1 (discussed below).

However, the situation with CO adsorption (Figure 3) is significantly different. While for PBE-D3BJ binding the energy of the first molecule is bigger than that of the second one, single points with other methods on this geometry show an opposite situation: the binding energy of the second molecule is twice (or even more) that of the first one. As it will be discussed further, a similar yet reduced effect was observed with high-level SPs on dicopper tetraacetate - carbon monoxide complexes. Looking at the sum of two adsorption energies, we end up on a picture qualitatively consistent with the previously discussed C_2_H_4_ adsorption:HF and CASSCF strongly underrate adsorption (even having first interaction in particular geometry repulsive);MP2, NEVPT2 and DCD-CAS(2) significantly overestimate the interactionl;MP3 and SOS-MP2 provide significant improvement to MP2 data and reaches the quality of single-reference coupled-cluster (or experimental data);DLPNO-MR-CCSD is 0.01 eV lower than DLPNO-CCSD (for the first adsorption).

DFT values show the higher distribution in comparison to C_2_H_4_ case and generally tend to overbind, having closest results obtained with the combination of B3LYP functional and D3BJ correction.

### 3.2. Dicopper Tetraacetate

A detailed insight into the performance of all above-listed methods can be obtained by looking into Table 1, where calculations for all 14 different molecular models for adsorption are summed up and compared with available experimental data from the literature. All molecular structures have been fully optimized with each DFT functional and confirmed that it is a local minimum by vibrational analysis. All systems have singlet ground state, except for the one adsorbed O_2_, which is triplet, and two oxygen molecules where the ground state is quintet. In the case of water as adsorbate, we can see that all three different types of DFT functionals are close to the experimental value, where GGA functional is the best in this case. From multi-reference methods, CASSCF reproduces PBE values while NEVPT2 overestimates binding energy for more than 0.1 eV/molecule. Both single-reference methods lie in between CASSCF and NEVPT2. Predicted adsorption energy of the second water molecule on the other apical position is up to 0.05 eV lower in all approaches. Carbon dioxide results differ in context that here higher-rung DFT functionals predict correct experimental value, while PBE slightly underestimate it. Here, CASSCF is the worse method, which underestimates the value almost by half, while NEVPT2 is producing correct binding energy comparable to B3LYP. Single-reference methods are again in between CASSCF and NEVPT2, DLPNO-CCSD(T) being better. The binding energy of the two CO_2_ is double that of the single molecule, telling us that there is no kind of cooperative effect between unsaturated Cu centers. Carbon monoxide has slightly higher binding energies (~0.05 eV higher on average) compared to carbon dioxide. All three DFT functionals exhibit similar behavior as CO_2_, again showing us that hybrid and meta-GGA functionals are slightly better. CASSCF method is erroneous, yielding negative binding energy for the first CO and just 0.04 eV for the binding of two CO molecules in total. From the other three ab initio methods, again NEVPT2 is the best, although all of them underestimate the value. It is interesting to notice that all DFT functionals slightly underestimate binding of the second CO while all other approaches overestimate it. The O_2_-adsorption is a very special case. While for other cases geometries of minima detected with PBE-D3 were found to be consistent with other approaches, O_2_-binding shows much more complex behavior, which was studied extensively (see Figure 4). Although the MP2 approach (used for high-spin state) normally predicts stronger binding and shorter intermolecular distance than DFT or CCSD, here we observe the opposite situation: MP2-minimum is significantly more shallow and located at a 0.5–0.7 Å higher distance than the PBE one. The DLPNO-CCSD significantly improves this result, but T2-diagnostic, that for all other systems lies in the range 0.068–0.070, in O_2_-PW-O_2_ system grows from 0.069 (on long-distance separation) to 0.1 at the geometry of DFT-minimum. In the authors’ opinion, this means that even for the high-spin state (six unpaired electrons for a particular system), closed-shell post-HF calculations have to be used with caution. The behavior of the NEVPT2 curve is even more unexpected: while on the long-distance region it shows stronger binding than MP2 (and generally overlaps with the CCSD curve), on the short distance it shows sharp growth, and on geometry of DFT-minimum binding energy calculated with this method it is even more negative than that of CASSCF. This is accompanied by significant slow-down and complication of CASSCF convergence (active space 4 × 4 for O_2_-PW and 6 × 6 for O_2_-PW-O_2_), despite active orbitals being significantly energetically separated from inactive ones, chemically intuitive and no flips between active and inactive space were observed. We speculate that such behavior might relate to intruder states in PT2 and for which even more advanced multi-reference treatment might be needed. Here we leave this part for a separate detailed study. All three DFT functionals slightly overestimate binding.

The end of the table is reserved for ethylene and two alcohols: methanol and ethanol. In the case of ethylene, all three DFT functionals produce results closed to the experiment with difference between functionals of only 0.01 eV. Both single-reference methods give the same result, in agreement with DFT. Multi-reference methods are the worst. CASSCF heavily underestimates the measured value while NEVPT2 overestimates it. The binding of the second ethylene molecule is not influenced by the presence of the first one on the other side of the Cu-PW. Predictions of geometry for both alcohols are similar. They adsorb though the -OH tail of the molecule. The binding energy of ethanol is slightly higher compared to methanol, which is in agreement with the experimental findings. All presented methods mostly overestimate the binding energies for more than 0.1 eV, although all three DFT functionals perform the best. All ab initio approaches are worse than DFT, CASSCF being the closest one to the measured values. In addition, it is interesting to notice that CASSCF overestimates the binding energy for methanol but underestimates it for ethanol.

## 4. Conclusions

Despite that single-reference post-HF methods are completely unreliable for description of the singlet state of systems of interest, they can be (for particular systems) straightforwardly used for getting adsorption energy in the triplet state. In fact, SOS-MP2, DLPNO-CCSD and DLPNO-CCSD(T) show results of experimental accuracy. All tested multi-reference approaches of reasonable cost appear to be of the same or lower quality than the tested DFT functionals. DLPNO-MR-CCSD benchmarks are very limited due to the huge cost, although the smallest possible active space was applied. With all written above we can specify the following points:In singlet state, DFT approaches perform fairly well despite the large and delocalized FOD and truly multi-reference nature of the system. This conclusion is significant for HKUST-1 calculations but should not be blindly extrapolated to other MOFs.CASSCF and NEVPT2 values can be applied to any kind of multi-reference case, but while the first underestimates the binding energies, the latter one overestimates them. Thus, these values should be used just as border criteria.We showed that the DLPNO-MR-CCSD approach can be used for the systems of this size, and we provide very trustful numbers, yet the cost of such calculations (especially with active orbitals on transition metals) makes them at the moment impractical.

To conclude this study, we observe that DFT has the most consistent performance, especially in the singlet state, being in almost all cases the method which is closest to the measured values. Slight advantages are functionals that belong to higher levels of Jacobs ladder. From ab-initio methods, DLPNO-(MR)-CCSD/(T) are the methods of choice, having in mind their computational cost compared to DFT.

## Figures and Tables

**Figure 1 molecules-27-00912-f001:**
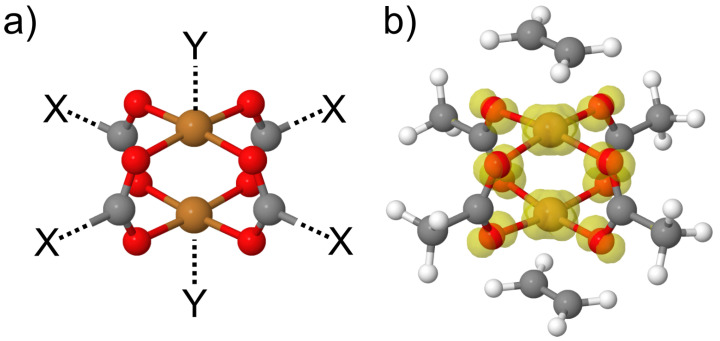
(**a**) Cu-dimer paddle-wheel model of metallic centers of HKUST–1. (**b**) Fractional Occupation Density of PW system with adsorbates. Isosurface value 0.005 e/Bohr^3^.

**Figure 2 molecules-27-00912-f002:**
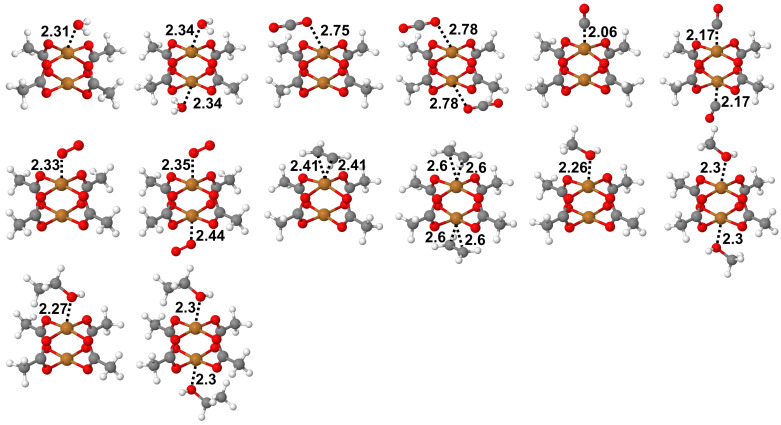
Structures of all 14 adsorbates on Cu-dimer-tetracarboxylate paddle-wheel obtained at PBE-D3BJ level of theory in Gaussian16. Local minima have been confirmed by analysis of vibrational modes.

**Figure 3 molecules-27-00912-f003:**
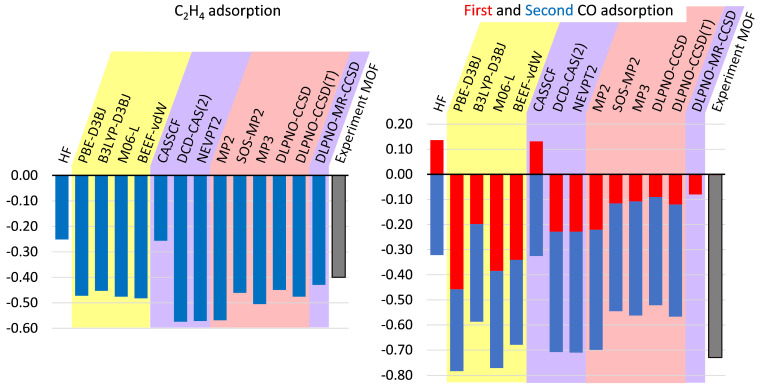
Adsorption energies (eV) of C_2_H_4_ and CO on dicopper tetraformate. Single points on PBE–D3BJ optimized geometries. Colors used to differentiate DFT (singlet state), single–reference post–HF methods (triplet state) and multi–reference approaches (averaged over singlet and triplet states).

**Figure 4 molecules-27-00912-f004:**
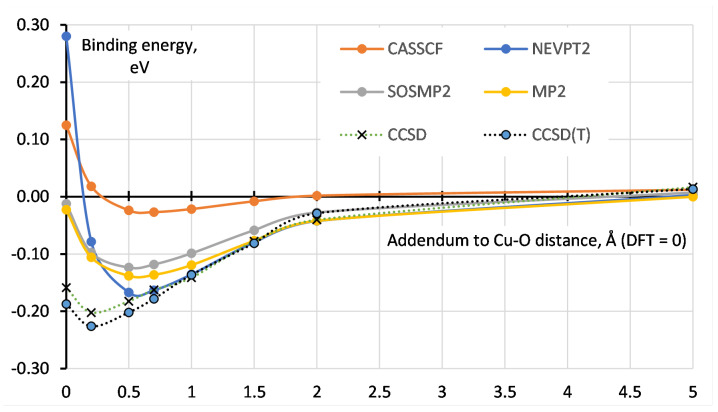
Rigid scans for the O_2_-PW-O_2_ system.

**Table 1 molecules-27-00912-t001:** The overview of calculated binding energies (in eV) of all educts and comparison with experimental values.

Adsorbate	PBE-D3 ${}^{a}$	B3LYP-D3 ${}^{a}$	M06-L	CASSCF	NEVPT2	SOS-MP2	CCSD(T) ${}^{b}$	Exp. E_b_ [Ref.]
H_2_O	0.54	0.58	0.59	0.53	0.65	0.60	0.62	0.53 ± 0.03 [45,89]
2H_2_O	1.05	1.14	1.16	1.03	1.27	1.17	1.21	
CO_2_	0.24	0.30	0.29	0.17	0.30	0.25	0.28	0.30 [63]
2CO_2_	0.48	0.60	0.57	0.34	0.59	0.49	0.57	
CO	0.41	0.33	0.39	−0.09	0.25	0.14	0.16	0.35–0.38 [90]
2CO	0.71	0.64	0.73	0.04	0.62	0.44	0.48	
O_2_	0.17	0.13	0.18	0.00 ${}^{c}$	0.08 ${}^{c}$	0.05 ${}^{c}$	0.11 ${}^{c}$	~0.1 [45]
2O_2_	0.31	0.25	0.33	0.03 ${}^{c}$	0.16 ${}^{c}$	0.12 ${}^{c}$	0.23 ${}^{c}$	
C_2_H_4_	0.43	0.44	0.44	0.21	0.54	0.43	0.45	0.35–0.37 [56], 0.40 [91]
2C_2_H_4_	0.78	0.83	0.81	0.29	1.14	0.86	0.94	
MeOH	0.55	0.55	0.60	0.56	0.70	0.66	0.66	0.42 [92]
2MeOH	1.06	1.20	1.14	1.05	1.36	1.26	1.28	
EtOH	0.60	0.69	0.66	0.40	0.90	0.75	0.82	0.48 [61]
2EtOH	1.15	1.34	1.26	0.89	1.63	1.40	1.51	

DFT results shown for the low-spin state, single-reference post-HF performed with high-spin state, multi-reference
approaches with averaging of both states. ^*a*^ D3BJ ^*b*^ DLPNO-CCSD(T) ^*c*^ energy calculated not in DFT geometry but
in the minimum of rigid scan, see Figure 4. Ref. [45] based on desorption at 60K; Ref. [61] from diffusion study,
weaker than water; Ref. [63] not an experimental value but GCMC-FF-DFT simulation; Ref. [90] estimated from
TPD studies.

## Data Availability

Additional data supporting this manuscript are available as Appendix A document from the publisher’s website or directly from the authors.

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
