# Peer review of "The Adsorption of Small Molecules on the Copper Paddle-Wheel: Influence of the Multi-Reference Ground State"

_molecules, 2022, doi:10.3390/molecules27030912_

Round 1

Reviewer 1 Report

This paper reports computationally calculated adsorption energies for small molecules on the Cu centers using various DFT and ab-initio methods. The reported calculations are extensive, and the authors used a variety of DFT functionals and Grimme dispersion corrections. I consider impressive that the authors used both molecular and periodic DFT software for their calculations. The paper is scientifically sound, appropriate for the journal, and mostly easy to read except for the overuse of acronyms as stated below in my comments. Moreover, the authors contained the needed bibliography. The paper should be accepted after the minor revisions given below:

  • The authors use several acronyms throughout the text, which make the reader hard to follow. For example, “HKUST-1”, which appears in the Introduction on page 2, has not been defined. The authors should minimize the use of acronyms in their text.
  • Figure 2. The X and Y letters need to be reduced in size.
  • Figure 3. The font size used is small. I suggest the authors reengage the information on this figure and put them in a column format instead of the current row format to allow larger font size.

Author Response

Reviewer 1

Comment: This paper reports computationally calculated adsorption energies for small molecules on the Cu centers using various DFT and ab-initio methods. The reported calculations are extensive, and the authors used a variety of DFT functionals and Grimme dispersion corrections. I consider impressive that the authors used both molecular and periodic DFT software for their calculations. The paper is scientifically sound, appropriate for the journal, and mostly easy to read except for the overuse of acronyms as stated below in my comments. Moreover, the authors contained the needed bibliography. The paper should be accepted after the minor revisions given below:

Author’s comment:  We thank Reviewer 1 for comments and suggestions to publish our manuscript after minor revision.

Comment: The authors use several acronyms throughout the text, which make the reader hard to follow. For example, “HKUST-1”, which appears in the Introduction on page 2, has not been defined. The authors should minimize the use of acronyms in their text.

Answer: We thank the Reviewer for the comment about acronyms. We are aware of the extensive use of them throughout the paper. At the end of the manuscript the complete list of all acronyms has been provided for the reference. We also consider that potential readers interested into this topic will already be familiar with most acronyms used in the manuscript. In the interest not to make a paper additionally significantly longer, we decide to keep the all the acronyms as they are.

Comment: Figure 2. The X and Y letters need to be reduced in size.

Answer: We have visibly reduced the size of the letters X and Y in Figure 1. The change can be seen in the attached document

Comment: Figure 3. The font size used is small. I suggest the authors reengage the information on this figure and put them in a column format instead of the current row format to allow larger font size.

Answer: We thank Reviewer 1 for pointing out the issue with the visibility of the labels in Figure 3. We have redrawn Figure 3 to present the same data in a way that labels can use a larger font for improved readability. The introduced change can be seen in the attached document.

Reviewer 2 Report

Report on the paper entitled " The adsorption of small molecules on the copper paddle-wheel:

Influence of the multi-reference ground state", by Marjan Krsti´c, Karin Fink and Dmitry Sharapa, submitted to Molecules.

The present paper studies the adsorption of seven molecules and seven couples of these molecules on the so-called paddle-wheel. This is a benchmark of a rather large set of methods, from standard DFT to sophisticated DFT methods, including dispersion correction, post Kohn-Sham corrections, (double hybrids), as well as single-ref and multireference post-Hartree-Fock methods. Appropriate software were used, with appropriate basis sets.

Methods, and structure of the 14 adsorbates on Cu-dimer-tetracarboxylate paddle-wheel are well described. A list of all the acronyms used in the paper, mainly those of all the quantum methods are given in (one page) table.

The paper is well written, including up to date references, and describes satisfactorily the work, which should be a reference for the selection of an appropriate method for studies of adsorption of (small) molecules over solid surfaces.

I recommend publication in the present form.

Author Response

Author’s comment: We thank Reviewer 2 for recognizing the quality of our work and recommendation for the publishing of our contribution.